# The Etiology and Risk Factors of Osgood–Schlatter Disease: A Systematic Review

**DOI:** 10.3390/children9060826

**Published:** 2022-06-02

**Authors:** Ludovico Lucenti, Marco Sapienza, Alessia Caldaci, Claudia de Cristo, Gianluca Testa, Vito Pavone

**Affiliations:** Department of General Surgery and Medical Surgical Specialties, Section of Orthopaedics and Traumatology, A.O.U.P. Policlinico Rodolico—San Marco, University of Catania, 95123 Catania, Italy; ludovico.lucenti@gmail.com (L.L.); alessia.c.92@hotmail.it (A.C.); decristo.claudia@gmail.com (C.d.C.); gianpavel@hotmail.com (G.T.); vitopavone@hotmail.com (V.P.)

**Keywords:** Osgood–Schlatter, pathogenesis, risk factors, etiology

## Abstract

The etiology and etiopathogenesis of Osgood–Schlatter Disease (OD) are not fully understood. The aim of this review is to systematically analyze the available literature about the etiology and risk factors of OD. The literature was systematically reviewed using the PRISMA criteria to evaluate all studies published in the last 25 years (between 1996 and 2021) dealing with the etiology of OD. A total of 16 articles were included. The etiology and risk factors of OD are controversial. The main articles focused on muscular factors (mainly tightness of the rectus femoris), alteration of the patellar tendon or extensor mechanism, mechanical factors (repetitive solicitation, trauma, sports), tibial anatomy (tibial slope or tibial torsion), and histological alteration. Associations with ankle kinematic and behavior disorders were also reported. Many theories about the etiology, risk factors, and associated factors of OD have been reported in the literature, but more studies are needed to fully understand the etiopathogenesis of this disorder.

## 1. Introduction

Osgood–Schlatter disease (OD) is a common pediatric disorder. It is a growth- and sports-associated knee injury with pain around the tibial tuberosity and morphology alterations around the apophysis (TTA) during adolescent growth. It often results from acute or chronic overload during sports activity, causing inflammation of the patellar tendon insertion on the tibial tuberosity [1]. It was first described in 1903 independently by Osgood in the US [2] and by Schlatter in Switzerland. Classically, the clinical presentation is associated with insidious onset (usually atraumatic) of anterior functional knee pain over the tibial tuberosity along with a bony prominence, as well as tenderness at the patellar tendon insertion site.

The pain usually occurs during and after physical activity and might be associated with local swelling. Many patients are completely asymptomatic, with less than 25% of patients reporting pain at the TTA [3,4,5]. The age of onset is typically between 10 and 15 years in boys and between 8 and 13 years in girls. Boys have it more often than girls, with a male-to-female ratio of 3:1 [3]. The prevalence of OD is 9.8%, and it can be bilateral in 20–30% of patients [4,5]. Many risk factors and activities have been associated with the increase of incidence of the pathology, but its etiology and etiopathogenesis are not fully understood.

OD is secondary to repetitive stress activities with the extensor mechanism [6]. However, this pathology is self-limited and resolves at the end stages of skeletal growth [7]. In most cases, symptoms usually resolve after the closure of the physis [6,7]. Sports commonly associated with OD are soccer, gymnastics, basketball, football, volleyball, and athletics [8]. In the differential diagnosis of OD, other diseases that need to be considered include Hoffa’s syndrome, Sinding–Larsen–Johansson syndrome, patellar tendon avulsion or rupture, soft tissue or bone tumor, chondromalacia patella, patellar tendinitis, accessory ossification centers, osteomyelitis of the proximal tibia, and tibial tubercle fracture [9].

The treatment is usually conservative, but some surgical options can be considered when conservative treatment fails or in some limited cases. Many studies have tried to analyze the etiology of OD over the years but have not fully clarified it. Therefore, the aim of this review is to investigate the available literature to provide an update on the evidence related to the etiology of OD and its risk factors.

## 2. Materials and Methods

This systematic review was conducted according to the guidelines of the Preferred Reporting Items for Systematic Reviews and Meta-Analyses (PRISMA) [10]. PROSPERO registration was not needed since it is a systematic review of literature. Two medical electronic databases (PubMed and Web of Science) were searched by a single author (LL) on 16 December 2021. The research string used was “(Osgood Schlatter OR Osgood-Schlatter) AND (pathology OR embryology OR etiology OR etiology OR etiopathogenesis OR genetics OR pathophysiology OR risk factors OR associated factors OR predisposing factors).” In total, *n* = 201 articles were found.

The initial titles and abstracts were screened using the following inclusion criteria: studies of any level of evidence reporting clinical or preclinical results published in the last 25 years (1996–2021) and dealing with the etiology of OD. All the articles written in languages other than English were excluded. All articles that dealt with different topics, had poor scientific methodology, or were without an accessible abstract were excluded. No duplicates were found (Figure 1).

## 3. Results

In total, 201 articles were found. At the end of the initial screening, we selected *n* = 117 articles that were eligible for reading in full text. After full-text reading, we ultimately selected *n* = 16 articles that satisfied the inclusion criteria. A PRISMA [11] flowchart of the selection and screening method is provided in Figure 1. Reference lists from the selected papers were also screened. The considered articles [12,13,14,15,16,17,18,19,20,21,22,23,24,25,26,27] report different factors associated with OD, which are summarized in Table 1.

### 3.1. Muscular Factors

Although no data are clear about the etiology of OD, muscular factors seem to have a clear role in the etiopathogenesis of the disease. Recently, Enomoto et al. [16] demonstrated that a more rigid rectus femoris in conditions of stretching with 45 and 90 degrees of flexion seems to be related to the onset of the disease. In addition, another three studies reported an association between tightness of the rectus femoris and OD [19,22,25].

### 3.2. Alteration of Patellar Tendon

Two different studies analyzed the correlation between the alteration of the patellar tendon and the onset of SD. Enomoto et al. [15] analyzed the elastic properties of the muscle–tendon unit in children with OD versus healthy children using ultrasound real-time tissue elastography. They analyzed 18 legs affected by OD and 42 legs in a control group. The strain ratio (SR) was used as an indicator of the elasticity of tissue. The SR of the patellar tendon in the OD group was significantly lower than in the control group (*p* < 0.05). On the other hand, there were no significant differences regarding the SR of the muscle group (*p* > 0.05). The results obtained suggested that a less elastic tendon is associated with a greater risk of OD development.

Visuri et al. [13] also analyzed the properties of the patellar tendon by comparing 82 knees of males with OD to 87 knees of healthy males who underwent confirmatory MRI. The study found that patients with OD had more elongated patellae and patellar tendons, possibly resulting from longstanding tension in the extensor apparatus on the knee during a growth spurt, when the femoral growth exceeds that of the knee structures.

### 3.3. Mechanical Factors

Seyfettinoğlu et al. [22] analyzed the degree to which anatomical alterations in patellofemoral alignment affect the onset of OD compared to repeated trauma. Two groups were analyzed. The first group included patients who were diagnosed with OD, and the second group included an equal number of age-matched patients with no known history of OD who came to the clinic with traumatic knee injuries and underwent a radiographic examination. The following characteristics were measured: Insall-Salvati (IS), Caton-Deschamps (CD), and Blackburne-Peel (BP) indices; congruence angles; femoral–patellar angles; sulcus angles; and patellar type according to the Grelsamer morphological classification. According to the results of the measurements, the main etiological factor appeared to be an increase in physical activity rather than variations in patellofemoral anatomy.

Gaulrapp et al. [24] analyzed how repetitive sports injuries could affect the onset of OD based on 126 patients. They concluded that OD primarily affects adolescent boys who are active in football and basketball and that it represents a structural response to repeated biomechanical stress. A history of repeated trauma underlying the etiopathogenesis of OD has also been confirmed by Abou El-Soud et al. [26] and De Lucena et al. [12].

### 3.4. Alteration of Tibial Anatomy

Alterations of the tibial anatomy also seem to be related to the etiopathogenesis of OD. Gigante et al. [17] analyzed 21 male patients with OD. It was found that an increase in external tibial torsion may play a key role in the onset of OD, especially in athletes. Green et al. [23] recently confirmed that OD is associated with increased posterior tibial slope (PTS).

### 3.5. Histological Alteration

Falciglia et al. [18] analyzed 13 patients with OD before surgery. They found that the fibrocartilaginous composition of the anterior tibial apophysis was altered in patients suffering from OD and that this could be the basis of the onset of the disease.

### 3.6. Other Factors

Guler et al. [21] analyzed the correlation between attention deficit/hyperactivity disorder (ADHD) and OD in collaboration with the psychiatric department. They found that 75% of the patients examined were affected by ADHD and that it was a risk factor for repeated trauma, which in turn represents a risk factor for OD. Finally, Sarcević et al. [27] found that limited ankle dorsiflexion is associated with increased knee flexion, tibial inversion, and foot pronation during the stance phase of running. These mechanisms could cause increased stress on the insertion of the quadriceps femoris muscle at the tibial tuberosity.

## 4. Discussion

The etiology of OD is still debated. Several conditions have been reported by different authors. Gigante et al. [17] used CT scans to evaluate 21 boys with OD and a control group of 20 adolescents, despite the fact that, in our opinion, the suspicion of OD does not warrant the use of CT scans. The results showed that the mean condylomalleolar and mean tibial torsion angles were higher in the OD group (<0.001) than the controls. They concluded that changes in the external tibial torsion angle and increased shear force (lateral–medial force) on tibial tuberosity are secondary to an increase in shear stress, especially during extension of the knee. This causes an increase in shear stress, which may play a role as a predisposing mechanical factor in the onset of OD in male athletes.

A study conducted in 2004 by Demirag et al. [14] evaluated 20 OD patients and a control group of 15 knees using MRI. They showed that the patellar tendon attachment is more proximal and broader above the tibial physis in the OD group, which according to the authors is likely to cause the pathology. Visuri et al. [13] looked at the relative length of the patellar articular surface, the length of the patella, and the Insall–Salvati, Blackburne–Peel, and Caton–Deschamps indexes. They evaluated the radiographic studies of 82 knees of 20-year-old males with OD and compared them to 87 healthy knees of 20-year-old male controls. This research demonstrated significant lengthening of the patellar body and an increased patellar height among the OD group compared to the controls.

In 2021, Enomoto et al. [16] compared the shear-wave velocity (SWV) of the rectus femoris and vastus lateralis during passive knee flexion and isometric contraction in 28 legs affected by OD and 26 legs without OD. Using shear-wave elastography, they observed a correlation between a stiffer rectus femoris under stretched conditions and OD. In 2020, the same authors evaluated the elasticity obtained by quadriceps muscles and patellar tendon in 18 legs affected by OD and 42 healthy legs using real-time tissue electrography [15]. The SR (the strain rate of the muscle or tendon divided by that of the reference material) was considered as an indicator of the elasticity of the tissue measured. The study revealed that the SR of the patellar tendon in the OD group was significantly lower than that in the control group, while there was no significant difference between the groups in terms of the SR value of all muscles.

Another study focused on muscles as a main risk factor for OD. In 2015, Nakase et al. [20] examined an OD group and a non-OD group during knee extension and found significant differences in quadriceps muscle tightness, body weight, and muscle tightness and strength. According to the authors, precise risk factors for OD were raised, particularly the tightness and strength of the quadriceps femoris muscle during knee extension and the flexibility of the hamstring muscles. The authors reported that quadriceps stretching may be a primary way to prevent OD.

The same authors evaluated 200 knees in 100 male football players aged 10–15 years [19]. In OD patients, increased quadriceps tightness and decreased hamstring tightness were not due to the femoral length during development of the tibial tuberosity alone. In addition, changes in thigh muscular tightness and performance of the quadriceps muscle may be involved with skeletal maturation of the distal patellar tendon insertion [19]. A cross-sectional analytical observational Brazilian study was conducted in 2011 on 956 adolescents [12]. The regular practice of sports in adolescents and the shortening of the rectus femoris muscle were found to be the principal causes associated with OD. Gaulrapp et al. [24] evaluated 126 patients with OD by conducting physical examinations, ultrasounds, and radiographic studies, and by recording participation in sports, growth rate, BMI, and muscle imbalance. Their results showed that only 10 out 126 patients did not practice any sports. They found that OD mainly affects teenage boys who are active in sports, especially in football and basketball, and represents a structural response to repeated biomechanical stress and joint overload. Age at onset, growth rate, BMI, and muscle imbalance did not seem to be predisposing factors. Some authors showed that there was a correlation between OD and history of trauma [26].

Green et al. [23] found an increased posterior tibial slope (PTS) in patients with OD compared to healthy knees. The reason for this finding was not completely clear. Stress from the extensor apparatus through the patellar tendon probably loads the anterior portion of the tibia excessively at the posterior segment, causing an asymmetric growth and increased PTS. A recent study by Seyfettinoğlu et al. [22] compared 40 patients with OD to 40 healthy patients and showed that the rate of patients engaged in sports activities was significantly greater in the OD group. According to the authors, the main etiological factor seemed to be massive physical activity rather than subtle variations of the patellofemoral anatomy and alignment of the extensor mechanism.

In a cohort study done by Watanabe et al. in 2018, 37 adolescent male soccer players were observed for 1 year, and measurements were recorded at baseline and every 6 months [25]. They collected information on physical function, the presence of Sever’s disease, and kicking motion. The study showed that pathogenic factors associated with OD in the support leg of patients included weight, height, BMI, bilateral tightness of the quadriceps femoris muscle, gastrocnemius muscle tightness, soleus muscle tightness, a diagnosis of Sever’s disease, and distance from the lateral malleolus of the support leg to the center of gravity during kicking.

However, some authors reported an alteration of the fibrocartilage tissue anterior to the ossification center, which seems to be altered before the action of tension forces. Falciglia et al. [18] did a histological study examining 13 specimens taken from 13 patients with OD before surgery. They hypothesized that the aspects of the fibrocartilage are the result of tissue weakness due to a physiopathological modification during pubertal growth.

Other studies focused on the relation between OD and the ankle joint. Sarcević et al. [27] found a high number of patients who were diagnosed with OD and had limited ankle dorsiflexion (less than 10°). The author proposed that these compensatory mechanisms could cause an increase in the mechanical stress exerted on the quadriceps muscle and in the insertion to the tuberosity of the tibia and represent a predisposing factor to the disease. Furthermore, an interesting study reported that ADHD is a significant risk factor for OD: 56 (75.6%) out of 74 children with OD examined had ADHD [21].

## 5. Conclusions

Many theories about the etiology, risk factors, and associated factors of OD have been reported in the literature. The most credible theories seem to focus on the variants of patellofemoral anatomy and alignment of the extensor mechanism, associated with overuse injuries. However, more studies are needed to understand the complex and multifactorial genesis of this osteochondrosis in order to reduce the risk factors and prevent the illness.

## Figures and Tables

**Figure 1 children-09-00826-f001:**
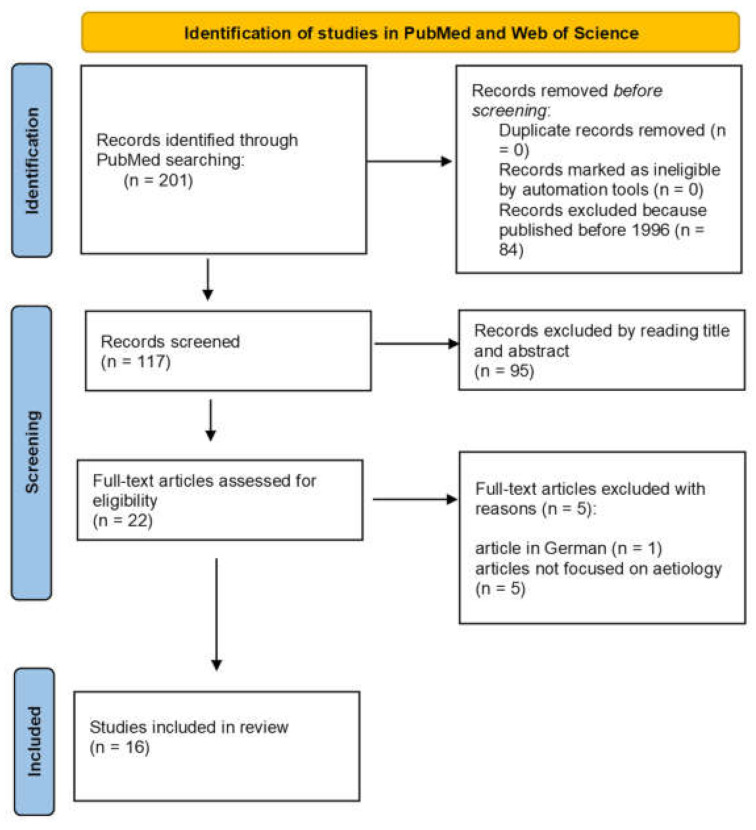
PRISMA flowchart.

**Table 1 children-09-00826-t001:** Studies of risk factors associated to OD.

Author	Year	Article	*n* of Patients with OD	*n* of Legs with OD	Control Group	Female	Male	Mean Age	Mean BMI	Risk Factor/Etiology/Association	Results	Level of Evidence	Type of Study
Enomoto	2021	Muscle stiffness of the rectus femoris and vastus lateralis in children with Osgood–Schlatter disease	28	28	26					A stiffer RF under stretched conditions (45° and 90° flexion) is related to the presence of OD.	RF and VL under unstretched and contracted conditions and the VL under stretched conditions have limited association with the presence of OD		Shear-wave velocity (SWV) of the RF and VL (in m/s) during passive knee flexion and isometric contraction measured using shear-wave elastography.
Gaulrapp	2021	The Osgood–Schlatter disease: A large clinical series with evaluation of risk factors, natural course, and outcomes	126	126		25	101	12.8	19.5	Repeated biomechanical stress	Adolescent boys practicing sports with repeated biomechanical stress. Age at onset, growth rate, BMI, and muscle imbalance are not significantly predisposing.	3	Prognostic study level III (longitudinal cohort study, consecutive patients without blinding, no control arm
Green	2020	Increased Posterior Tibial Slope in Patients with Osgood-Schlatter Disease: A New Association	40	38	32	20	20	12.6		Increased posterior tibial slope (PTS)	An association between OD and increased PTS.	3	Radiographic evaluation comparing PTS in 40 knees with OD and 32 control knees
Enomoto	2020	The Passive Mechanical Properties of Muscles and Tendons in Children Affected by Osgood–Schlatter Disease	18	18	42	0	18	13.6		Low elasticity of Patella tendon (PT)	Patella tendon (PT) with a lower SR associated with OD. The passive mechanical propertiesof the quadriceps muscles have limited association with an OSD	4	Elasticity obtained from the quadriceps muscles and patella tendon (PT), using real-time tissue elastograph, in 18 legs affected by OD and 42 healthy legs
Seyfettinoğlu	2018	Is There a Relationship between Patellofemoral Alignment and Osgood–Schlatter Disease? A Case-Control Study	40	40	40	10	30	12.88	19.5	Increased physical activity	The main etiologic factor seems to be increased physical activity rather than subtle variations in patellofemoral anatomy and alignment of extensor mechanism.		Prospective observational case-control study conducted on two groups of adolescent patients–radiographic evaluation
Watanabe	2018	Pathogenic Factors Associated with Osgood–Schlatter Disease in Adolescent Male Soccer Players: A Prospective Cohort Study	12		24	0	12	10.2	17.1	A diagnosis of Sever disease and backward shifting of the center of gravity during kicking, quadriceps femoris muscle tightness bilaterally, gastrocnemius muscle tightness, soleus muscle tightness	Developmental stage, physical attributes, preexisting apophysitis. Sever disease and backward shifting of the center of gravity during kicking intensified the risk of OD.	2	Prospective Cohort Study Variable evaluated: morphometry, joint flexibility, lower extremity alignment, presence of Sever disease, kicking motion
Nakase	2015	Precise risk factors for Osgood–Schlatter disease		10	60	0	10	12.6	19.8	The quadriceps femoris muscle tightness and muscle strength during knee extension and flexibility of the hamstring muscles	Tightness and strength of the quadriceps femoris muscle during knee extension and flexibility of the hamstring muscles.		Prospective cohort study
Guler	2013	Is there a relationship between attention deficit/hyperactivity disorder and Osgood–Schlatter disease?	74			0	74	12.64		ADHD—attention deficit/hyperactivity disorder	Strong association between OD and ADHD	2	Prospective study
Nakase	2014	Relationship between the skeletal maturation of the distal attachment of the patellar tendon and physical features in preadolescent male football players	100			0	200	12.0		Increased quadriceps tightness with rapidly increasing femoral length during tibial tuberosity development	Quadriceps tightness increased, hamstring tightness decreased, suggesting that quadriceps tightness is not due to femoral length alone but that muscle strength may also be involved.	3	Cross-sectional study, Level III.
Falciglia	2011	Osgood Schlatter lesion: histologic features of slipped anterior tibial tubercle	13							The fibrocartilage anterior to the ossification centre	The slippage of the patellar tendon insertion may be progressive and caused by pathological fibrocartilage.		Histology
Amany M. Abou El- Soud	2010	Prevalence of osteochondritis among preparatory and primary school children in an Egyptian governorate	14	15						History of Trauma			Cross-sectional study
de Lucena	2011	Prevalence and associated factors of Osgood-Schlatter syndrome in a population-based sample of Brazilian adolescents	94			40	54			Regular practice of sports and shortening of Rectus femoris	Regular practice of sports in adolescents and the shortening of the rectus femoris muscle were the main factors associated with the presence of OD	3	cross-sectional analytic observational study
Demirag	2004	The pathophysiology of Osgood-Schlatter disease: a magnetic resonance investigation	20		15			13.4		Patellar tendon attachment more proximal and broader above the tibial physis	If the patellar tendon attaches more proximally and in a broader area to the tibia, this might cause OD		MRI study
Gigante	2003	Increased external tibial torsion in Osgood-Schlatter disease	21		20	0	21	14		Increase in external tibial torsion	Increase in external tibial torsion may be a predisposing mechanical factor in the onset of OD.		CT scan evaluation
Visuri	2007	Elongated patellae at the final stage of Osgood-Schlatter disease: A radiographic study	82		87			20.7	23.4	Increased patellar height, elongated patellae and patellar tendons	Elongated patellae and patellar tendons which may result from longstanding tension of the extensor apparatus during growth spurt, when femoral growth exceeds that of the anterior structures of the knee		X-ray
Sarcević	2008	Limited ankle dorsiflexion: a predisposing factor to Morbus Osgood Schlatter?	45			5	40			Limited dorsiflexion of the ankle	Limited dorsiflexion of the ankle joint might be important for developing OD		Clinical evaluation–ROM of the ankle

## Data Availability

Not applicable.

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
