# Peer review of "The Etiology and Risk Factors of Osgood–Schlatter Disease: A Systematic Review"

_children, 2022, doi:10.3390/children9060826_

Round 1

Reviewer 1 Report

History, clinical presentation, epidemiology and treatment of Osgood-Schlatter disease are described in detail in Introduction. Since the ethology and risk factors for Osgood-Schlatter disease are still controversial there is a need for a systematic review to summarize currents opinions on these issues. 

Systematic review was conducted according to PRISMA guidelines. All selected articles are presented in detail in Result section and properly discussed in Discussion section.

This paper summarizes current knowledge on the etiology of Osgood-Schlatter disease and would be useful to clinicians and others interested in this disease. 

Author Response

Dear Author,

Thank you for your valuable contribution.

I did the revisions as requested. Specifically I modified the two sentences as you requested, I added to discussion and coclusione that OD is classified as an overuse-injury and finally I revised the English language.

Reviewer 2 Report

Since this review discusses the ethiology of OD, this sentence should be revises "It results from acute or chronic overload during adolescent sports, which causes inflammation of the patellar tendon insertion on the tibial tuberosity [1]"

Regarding "Gigante et al. [17] used CT scans to evaluate 21 boys with OD and a control group of 20 adolescents" it should be noted, that suspicion of OD does not legitimate the use of CT scans.

OD is classified as an overuse-injury, this should be mentioned in the discussion and conclusion.

Author Response

(The authors gave the same response as above.)
